# Creativity: The Effectiveness of Teacher–Student Conflict

**DOI:** 10.3390/ijerph19031628

**Published:** 2022-01-31

**Authors:** Ruoying Xie, Jinzhang Jiang

**Affiliations:** 1School of Media & Communication, Shanghai Jiao Tong University, Shanghai 200240, China; yuraxie@sjtu.edu.cn; 2USC-SJTU Institute of Cultural and Creative Industry, Shanghai Jiao Tong University, Shanghai 200240, China

**Keywords:** students’ creativity, teacher–student understanding conflict, teacher–student process conflict, teacher–student relationship conflict, sustainable development education

## Abstract

This study examines the effectiveness of different types of teacher–student conflict in promoting students’ creativity in universities. Previous studies mainly focused on the negative effects of conflict; few examined its positive effects. Teacher–student conflict in university classes can take many forms; however, there are no clear definitions of the various types of such conflict. This study classified teacher–student conflict as understanding conflict, process conflict, and relationship conflict, and we used this classification to extend prior research by revealing the beneficial impacts of teacher–student conflict on students’ creativity. We empirically examined the relationship between teacher–student conflict and students’ creativity. The hypotheses were supported by using data from questionnaires completed by 2009 students at 17 American universities. We then conducted a hierarchical regression analysis of the data using structural equation modeling. The findings indicate that understanding conflict and process conflict had significant positive effects on students’ creativity, whereas relationship conflict had a significant negative effect on students’ creativity. This study thus revealed the positive effect of teacher–student conflict in university classes and suggests encouraging conflict (understanding conflict and process conflict) as a unique teaching method to stimulate students’ creativity.

## 1. Introduction

In the 2030 Agenda for Sustainable Development of the United Nations, sustainable development education is one of the 17 sustainable development goals [1]. Creativity is a key driver for sustainable social development and an important guarantee for promoting sustainable development [2,3]. In sustainable development education, enhancing student creativity was also emphasized [1]. As the crucial part of the educational system, the responsibility of the university includes not only disseminating knowledge but also cultivating student creativity [4,5].

Creativity is generally defined as novel, useful products [6]. Students’ novel, useful products in the classroom are new ideas that are put forward by themselves. To promote students’ creativity, researchers recommended encouraging students to ask more questions rather than merely providing them with the answers [7,8,9]. However, students put forward questions or disagreements that may lead to teacher–student conflict. Previous studies concluded that conflict was negative and can result in a broad range of social and academic problems, such as misconduct, disciplinary infractions, and school suspensions [10,11,12]. As a result, the researchers suggested avoiding and eliminating all teacher–student conflicts [11,13]. Hence, conflict’s beneficial role has rarely been studied.

Conflicts should not always be regarded as negative behavior. Conflict is the disagreement in opinions between people or groups [14,15,16]. The occurrence of conflict in the classroom is uncontrollable and unpredictable [17]. Previous research emphasized encouraging students’ competence to solve uncertainty for enhancing students’ creativity. In this regard, the ability of how to deal with conflict should be reconsidered. Students’ creativity requires challenging ideas and critical thinking, and teacher–student conflict leads to healthy debate and productive controversy, which expands students’ perspectives and promotes divergent thinking [7]. Consequently, we predict that teacher–student conflict can benefit students’ creativity.

Rather than seeking how to eliminate or avoid teacher–student conflict in the classroom, students’ creativity provides a new perspective on the positive effects of conflict. Therefore, this study has three objectives: (1) we aim to describe and explain the definition of teacher–student conflict in the university classroom; (2) the types of teacher–student conflict are ambiguous; in order to research the positive effect of conflict, we classify three different types of teacher–student conflict in the classroom (understanding conflict, process conflict, and relationship conflict); (3) we aim to determine how different types of teacher–student conflict affect students’ creativity.

## 2. Theoretical Background and Literature Review

### 2.1. Conflict Theory and Creativity

Previous research mainly focused on teacher–student relationship conflict [17,18] but seldomly mentioned other types of teacher–student conflict. Several studies consistently concluded that relationship conflict is not conducive to students’ creativity [18,19]. Nevertheless, are all types of conflict negative? This paper will research all types of conflict in one theoretical model to explore their effects.

Creativity is a critical aspect of addressing sustainability [20,21,22,23,24]; it is the key skill to achieve sustainable development and foster more sustainable societies [25,26]. Creativity assists us in gaining clarity of vision and accessing the breakthrough thinking required for sustainable development [27]. Researchers have scrutinized students’ creativity in relation to a variety of other constructs, such as antecedents, mediators, and moderators [5,28,29,30]. However, sustainable development education is an ongoing process that emphasizes asking questions rather than accepting unchangeable answers [31]. Creativity in sustainable development education necessitates the development of personal consciousness rather than absorbing prepackaged knowledge, conformity, and a limited intellectual scope [7]. Students’ creativity is the requirement of challenging behavior and critical thinking, especially in uncertain and ambiguous circumstances. In the classroom, teacher–student conflict is unforeseeable and uncontrollable. Therefore, it is crucial to investigate the relationship between teacher–student conflict and students’ creativity.

### 2.2. Understanding Conflict, Process Conflict, and Relationship Conflict

It is necessary to clarify and explain the various types of teacher–student conflict in the classroom. Teacher–student conflict has not been demonstrated systematically, and the classifications and definitions of different types remain indistinct.

Conflict in the organizational context has been categorized as task conflict, process conflict, and relationship conflict [16,32,33]. In organizations, different types of conflict have different effects on how individuals perceive disagreements regarding tasks, work processes, and relationship issues [32,34]. Conflict theory in organization studies has been empirically demonstrated, which provides theoretical support to categorize and explain teacher–student conflicts in the university classroom. The concepts of teacher–student conflicts and organizational conflicts are explained correspondingly.

In the workplace, task conflict entails disagreements about the content and outcomes of the team task [15,16]. In the classroom, the teacher’s main task is to teach knowledge, and students’ main task is to learn knowledge. Because of teacher–student differences in experience, cognition, and other aspects, their understanding of knowledge may be inconsistent [35]. Differing views on knowledge understanding can be considered as understanding conflict.

Process conflict involves disagreements concerning how to accomplish a task or project [36]. In the classroom, disagreements can occur regarding the logistics of how to teach and learn. Teachers use a wide range of teaching approaches [35], and students apply a variety of learning styles [37]. In the process of imparting knowledge, the mismatch between students’ learning styles and teachers’ teaching styles is process conflict.

Relationship conflict involves disagreements related to interpersonal emotional incompatibilities, such as personality differences, norm differences, and value differences [38]. In the classroom, differing personalities and values induce teacher–student relationship conflict [39].

## 3. Hypothesis Development

### 3.1. The Relationship between Understanding Conflict and Students’ Creativity

In the classroom, the products of students’ creativity are their new ideas. The differing opinions on knowledge understanding between teacher and student induce understanding conflict [32]. Understanding conflict emerges as a consequence of differences in cognition and knowledge background, but the difference may help students bring new perspectives to the class. Throughout the discussion on various points of view and ideas, based on understanding conflict, students may come up with fresh solutions to the problem. Understanding conflict can also provide students a deeper understanding of knowledge during the discussion, which enables more opinions to be shared and promotes information exchange. The right or wrong of the objection is not important; the behavior of asking questions is critical thinking, which can improve students’ creativity [7]. Therefore, we propose Hypothesis 1.

**Hypothesis** **1.***In the classroom, teacher–student understanding conflict is positively correlated with students’ creativity*.

### 3.2. The Relationship between Process Conflict and Students’ Creativity

In addition to understanding conflict, incompatibilities or differences in learning styles and teaching styles also induce conflict: process conflict. Students’ learning style is how they receive and process knowledge in learning situations [37]. Learning styles have been categorized in various ways; for example, Kolb’s model [40] focuses on how knowledge is grasped, and Labib et al. [37] focus on visual and auditory perception. As for the teaching style, it reflects teachers’ attitudes toward philosophy, beliefs, and values between their teaching and practice [35]. The teaching style is shaped by teachers’ personal teaching experience, educational background, and cultural background [35]. Previous studies have shown that well-matched teaching and learning styles are effective in transferring knowledge to students [39,41,42]. Additionally, the researchers [39,41,43] have offered recommendations on what teaching styles are appropriate for certain learning styles.

However, a mismatched teaching and learning style is not always detrimental for the class.Teachers and students communicate in the classroom based on the interaction of their teaching styles and learning styles. Because the interactions are constantly changing and cannot be easily controlled [41], both the teaching style and learning style are not easy to perfectly match. Therefore, it is challenging to find a combination of teaching and learning styles that is effective during the whole process of one class [41]. Furthermore, even if the learning and teaching styles are initially well-matched, this state may not always persist. As a result, it is time to consider whether matching them is worthwhile.

Process conflict may be effective in this regard. We propose that unmatched teaching and learning styles can enhance students’ creativity. For example, the teacher chooses to demonstrate knowledge through theory, while the student expects a case study. Process conflict provides a chance for both teachers and students to explore the rationale for their style choices, which helps them understand each other’s requirements. In this way, process conflict may lead to their style adaptation or the creation of new teaching styles or learning styles.

In contrast, in a classroom with less process conflict, students are more likely to be required to fit an established teaching style [42]. In such cases, teachers may strictly adhere to their planned teaching style until the class ends. They may not consider students’ feedback, answer students’ objections, or allow students to ask challenging questions. The creation of students’ new learning styles is impeded.

Based on these discussions, we propose Hypothesis 2.

**Hypothesis** **2.***In the classroom, teacher–student process conflict is positively correlated with students’ creativity*.

### 3.3. The Relationship between Relationship Conflict and Students’ Creativity

Differences in personalities and values can induce relationship conflict. Relationship conflict is related to interpersonal tension, which induces jealousy, worry, or anger [15]. Hostile or rebellious feelings between teachers and students reduce students’ involvement in the classroom [19]. In such cases, both the teacher and the student are likely to focus on each other as adversaries rather than on solving knowledge problems [11].

Relationship conflict can trigger two types of extreme behaviors: overreaction and excessive silence [43]. Firstly, Myers & Cowie [43] concluded that overreactions would interfere with classes. Teachers’ aggressive behaviors, for example, interrupting students’ talking, ignoring their questions, and ridiculing them, might impede the teaching process and inhibit the achievement of the teaching purpose [39]. In addition, the overreaction of students, such as threats and personal attacks, also disrupts the class, which decreases the effectiveness of teaching and reduces other students’ learning opportunities [19].

Secondly, excessive silence resulting from relationship conflict is detrimental to students’ creativity. Excessive silence manifests as non-response and evasion in the face of teachers’ questions. Even if they are interested in the teacher’s question, students choose not to participate in the class process [11].

Based on the literature, we propose Hypothesis 3.

**Hypothesis** **3.***In the classroom, teacher–student relationship conflict is negatively correlated with students’ creativity*.

## 4. Materials and Methods

### 4.1. Data Collection Procedure

A total of 2027 participants were included in the analysis, with data from 18 students eliminated due to age non-compliance target (i.e., Ph.D. students). The final sample included 2009 students (50.37% females) ranging from 18 to 23 years old (M = 20.71, SD = 1.58) from 17 American universities. The research was approved by institutional review boards and education authorities. Students were recruited via email and bulletin board announcements. Confidentiality and anonymity were ensured, and all participation was voluntary and anonymous. Each participant was entered into a random drawing to receive a coffee coupon valued at USD 20 as a token of appreciation.

At the top of each page, a clear explanation was provided to ensure that the participants were aware of how to respond. An example is “Please recall the experience in the classroom. There is no right or wrong answer. Give your choice without too much consideration”. After the participants had answered all the questions, they entered their demographic information to qualify for the drawing. Their identifying information was kept separate from the data analysis.

In order to suit the context of this study, we conducted in-depth interviews. For example, to measure the students’ creativity, we used a questionnaire from the Creative Achievement Questionnaire (CAQ) [44]. An example from the original questionnaire is “I designed an invention for a course assignment and sold it to someone I knew”. However, the results of interviews indicated that the probability of this phenomenon is low (1.3%). The reason may be that the students who were tested with the original CAQ were all from Harvard University, whereas this study measured students’ creativity on average. As a result, we deleted this item.

In addition, the expression of the questionnaire was also modified through in-depth interviews with teachers and students. For example, pilot tests revealed that 7.9% of students chose “strongly agree” or “agree” in response to the question “Do you have conflicts with teachers in the classroom during this semester?” However, the in-depth interviews indicated that conflict often occurred (68.2%). We found the reason may be that students thought only intense disagreements (e.g., quarrels) could be defined as a conflict. Hence, we added an explanation (“disagreement between the teacher and students are considered as a conflict”) in the final version of the questionnaire. This change increased the validity of the questionnaire from 0.79 to 0.88.

### 4.2. Measurement Instruments

The responses to the items were given on a Likert scale, with responses ranging from 1 (“strongly disagree”) to 5 (“strongly agree”). The details of scales used in this study are presented as follows.

#### 4.2.1. Students’ Creativity

It was assessed by Creative Achievement Questionnaire [44], which included 10 domains with 61 questions, such as visual arts, music, and inventions. A sample item is “I wrote an original short story or poem for a class”.

#### 4.2.2. Understanding Conflict

It was assessed using the scale developed by Cronin and Weingart [45]; a sample item is “Teacher and I frequently have disagreements about the understanding of the content in the classroom”.

#### 4.2.3. Process Conflict

It was assessed using the scale adopted by Cronin and Weingart [45]; for example, “I frequently have disagreements about teacher’ teaching styles in the class”.

#### 4.2.4. Relationship Conflict

It was assessed using the scale developed by Jehn and Mannix [36]; a sample item is “My teacher and I had quite different personalities”.

## 5. Data Analysis and Results

As can be seen in Table 1, the majority were female students (50.37%) of the total number of 2009 responses. Regarding age, there is a certain balance between the four grades. In terms of major, the percentage of social science, art, and business accounts for 79.95%.

Table 2 reports the means, standard deviations, and inter-scale correlations between all the study variables. Understanding conflict and process conflict were both positively correlated with students’ creativity (r = 0.870, *p* < 0.01; r = 0.570, *p* < 0.01, respectively). Relationship conflict negatively predicted students’ creativity (r = −0.590, *p* < 0.01).

The results of the correlation analysis revealed that all the variables had significant associations, supporting the use of regression and mediating analysis in the following steps.

The results of the hierarchical regression shown in Table 3 (M2) support Hypotheses 1–3. Understanding conflict and process conflict had significant positive effects on students’ creativity (β = 0.019, *p* < 0.01; β = 0.141, *p* < 0.01, respectively). Relationship conflict was significantly negatively related to students’ creativity (β = −0.505, *p* < 0.01).

## 6. Discussion

Creativity offers a unique perspective on conflict’s beneficial impacts. The results indicated that understanding conflict and process conflict increased students’ creativity, but conflict involving negative emotional personal attacks did not.

Regarding the first goal, we described and explained the teacher–student conflict in the classroom. For the second goal, we outlined an evident classification of different types of teacher–student conflicts. For the third goal (determining how different types of teacher–student conflict affect students’ creativity), the results revealed that understanding conflict and process conflict were positively related to students’ creativity (r = 0.870, *p* < 0.01; r = 0.570, *p* < 0.01, respectively).

Our findings show that students can improve their creativity by putting forward different ideas during conflicts rather than merely accepting teachers’ perspectives. Understanding conflict stimulates discussion and encourages knowledge sharing and prevents students’ excessive conformity with teachers, which can improve students’ creativity. Process conflict facilitates new teaching styles and learning styles, which also benefits students’ creativity.

On the contrary, the results indicated that relationship conflict was negatively correlated with students’ creativity (r = −0.590, *p* < 0.01). Relationship conflict evokes negative emotions and amplifies aggression [13,46,47]. It provokes oppositional behavior regarding personal issues, hostile reactions, and heightens interpersonal tension. Relationship conflict results in a discussion that is off topic from class content and easily escalates into personal conflict. Thus, relationship conflict reduces students’ creativity. These findings expand those of studies about the negative effect of relationship conflict [11,18,46,47].

Our research makes several contributions to the research of conflict and creativity. Firstly, few studies have systemically assessed all types of teacher–student conflict in the classroom. Based on conflict theory [14,48,49,50]), this study classified and explained three types of teacher–student conflict. Secondly, we investigated how different types of conflict are associated with students’ creativity. In contrast with previous studies, which have primarily focused on the detrimental consequences of conflict [10,49,51], this research provided empirical evidence regarding the positive outcomes of two types of teacher–student conflict in the classroom (understanding conflict and process conflict).

These findings have important implications for practitioners of sustainable development education. We suggested looking at conflict as a chance to improve students’ creativity. For teachers to effectively transmit knowledge and skill, it is necessary to reduce or eliminate relationship conflict in the classroom [43]. However, because critical thinking was highlighted in students’ creativity, conflict should not always be considered as a disruptive factor in the regular class process. Many teachers are unaware of the potential for conflict, seek to prohibit it, or deal with it irrationally [43,51]. A more effective teacher is able not only to convey knowledge but is also dedicated to inspiring students’ creativity. Our recommendations for practice are, therefore, as follows: (1) teachers should have the ability of deviating from a fixed process and deal with unexpected conflict by taking conflict as an opportunity to cultivate students’ creativity; (2) teachers should guide discussion centered on course content to foster understanding conflict and process conflict; (3) teachers should seek to prevent relationship conflict and make resolving it their primary priority when it is triggered in the classroom.

This study has a few limitations that present opportunities for future research. Firstly, we did not consider virtual classes, which are becoming increasingly common. Secondly, creativity is a lifelong behavior that requires longer observation periods for clearer and more objective identification. However, this study did not assess students’ long-term behavior (over 1 to 3 years) after being involved in the conflict. Future studies could use this conflict–creativity model in other countries or cultures to generate comparative studies. We would also welcome further studies on the impact of teacher–student conflict on teachers’ creativity, and on how other students who are not directly involved in the conflict are motivated. Additionally, it would be interesting to investigate the same hypothesis in European universities as well.

## 7. Conclusions

The findings of this study show how various types of conflict impact students’ creativity. Previous studies have shown that teachers focus on resolving or avoiding conflict, and their attitudes are not inclusive when challenged with a conflict. However, in light of the positive aspects of understanding conflict and processing conflict, teachers’ beliefs about conflict inclusion should be strengthened. This will not only aid in the occurrence of more beneficial conflicts but will also benefit students’ creativity. The teachers’ inclusion should not be overlooked because it reveals the negative side of relationship conflict. Teachers’ tolerance and concern for students can contribute to the enhancement of the interpersonal relationships between teachers and students, which can reduce the negative effect of relationship conflict.

## Figures and Tables

**Table 1 ijerph-19-01628-t001:** Sample characteristics.

	N	Percentage
Gender		
Male	908	45.19%
Female	1012	50.37%
Others	89	4.44%
Grade		
Freshman	487	24.24%
Sophomore	502	24.98%
Junior	506	25.18%
Senior	514	25.60%
Major		
Social Science	557	27.72%
Business	523	26.03%
Science	403	20.05%
Arts	526	27.20%

**Table 2 ijerph-19-01628-t002:** Means, standard deviations, and correlations.

Variable	M	SD	1	2	3	4	5	6
1. Gender	1.15	0.772	1					
2. Grade	2.35	0.883	0.564 **	1				
3. UC	2.08	0.694	0.574 **	0.678	1			
4. PC	2.44	0.821	0.029 **	0.242 **	0.233	1		
5. RC	2.75	0.643	0.311 **	0.277 **	0.355 **	0.444	1	
6. SSC	2.99	0.782	0.796	0.490 **	0.870 **	0.570 **	−0.590 **	1

Note: ** *p* < 0.01 (two-tailed); UC = understanding conflict; PC = process conflict; RC = relationship conflict; SSC = students’ creativity.

**Table 3 ijerph-19-01628-t003:** Hierarchical regression analysis.

Variable	Students’ Creativity	
	M1	M2
Gender	0.088	0.046
Grade	0.167	0.124
Understanding Conflict		0.019 **
Process Conflict		0.141 **
Relationship Conflict		−0.505 **
R2	0.161	0.443
∆R2	0.162	0.282
F	26.261 ***	97.451 ***

Note: ** *p* < 0.01; *** *p* < 0.001 (two-tailed).

## Data Availability

The data presented in this study are available on request from the corresponding author.

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
