# Peer review of "Creativity: The Effectiveness of Teacher–Student Conflict"

_ijerph, 2022, doi:10.3390/ijerph19031628_

Round 1

Reviewer 1 Report

Interestingly chosen standpoint and analysis of the conflict between the teacher and the student. As mentioned by the authors, this phenomenon is very often analysed as a negative factor, but this article reveals its positive sides.

The key thing that I miss in this article is a more thorough rationale for SUSTAINABLE creativity and the disclosure of the elements of its content. I think the authors should describe this concept more clearly by revealing its content and elements. Otherwise, the question arises as to whether this is not an artificially chosen concept which merely gives an imaginary impression of compatibility with the journal’s topics.

In this case, the question also arises as to the employed research instrument for assessment of sustainable creativity. The impression is that the questionnaire that is used is intended simply to measure creativity, without considering its specific type, namely sustainable creativity. The connectedness of this instrument and sustainable creativity needs to be substantiated.

In a general sense, the results of the article are relevant and interesting but they are more reflective of the educational impact and links; therefore, the connectedness of this publication and the journal’s philosophy and ideas; i.e., Good-Health and Well-Being, Education and Social Inclusion, is questionable. The part of the discussion should be supplemented by proving that the obtained results pertain to these phenomena and how they relate to them.

Author Response

We appreciate your careful reading of our manuscript and the constructive comments. We apologized for omitting these critical problems and have made all revisions in response to your valuable suggestions. And we are pleased to listen to your advice again if there is anything that does not meet your standards.

Point 1: The key thing that I miss in this article is a more thorough rationale for SUSTAINABLE creativity and the disclosure of the elements of its content. I think the authors should describe this concept more clearly by revealing its content and elements. Otherwise, the question arises as to whether this is not an artificially chosen concept which merely gives an imaginary impression of compatibility with the journal’s topics.

Response 1:

We totally agree that the disclosure of sustainable creativity is lacking, and most of our literature on creativity is not related to sustainable creativity. The theory and elements about sustainable creativity were seldom, and all the research was present in the last manuscript. As you questioned, we mentioned more about the general students’ creativity. Hence, we choose to use the original classic students’ creativity concept instead of the immature theory of sustainable creativity. Promoting students’ creativity is also an important issue in sustainable development education. The theme of this special issue is concerned with sustainable development goals, sustainable development education is one of the 17 sustainable development goals [1]. In this regard, the intro and literature are updated as follows.

Revised Contents:

1. Introduction

In the 2030 Agenda for Sustainable Development of the United Nations, sustainable development education is one of the 17 sustainable development goals [1]. Creativity is a key driver for sustainable social development and an important guarantee for promoting sustainable development [2, 3]. In sustainable development education, enhancing student creativity was also emphasized [1]. As the crucial part of the educational system, the responsibility of the university includes not only disseminating knowledge but also cultivating student creativity [4, 5].

Creativity is generally defined as the novel, useful products [6]. Students’ novel, useful products in the classroom are new ideas which put forward by themselves. To promote students’ creativity, researchers recommended encouraging students to ask more questions rather than merely providing them with the answers [7-9]. However, students put forward questions or disagreements that may lead to teacher-student conflict. Previous studies concluded that conflict was negative and can result in a broad range of social and academic problems, such as misconduct, disciplinary infractions, and school suspensions [10-12]. As a result, the researchers suggested avoiding and eliminating all teacher-student conflicts [11, 13]. Hence, conflict's beneficial role has rarely been studied.

 Conflicts should not be always regarded as negative behavior. Conflict is the disagreement in opinions between people or groups [14-16]. The occurrence of conflict in the class is uncontrollable and unpredictable [17]. Previous research emphasized encouraging students’ competence to solve uncertainty for enhancing students’ creativity. In this regard, the ability how to deal with conflict should be reconsidered. Students’ creativity requires challenging ideas and critical thinking, teacher-student conflict leads to healthy debate and productive controversy, which expands students’ perspectives and promotes divergent thinking [7]. Consequently, we predict that teacher-student conflict can benefit students’ creativity.

Rather than seeking how to eliminate or avoid teacher-student conflict in the classroom, students’ creativity provides a new perspective on the positive effects of conflict. Therefore, this study has three objectives: (1) We aim to describe and explain the definition of teacher-student conflict in the university classroom; (2) The types of teacher-student conflict are ambiguous, in order to research the positive effect of conflict, we classify three different types of teacher-student conflict in the classroom (Understanding Conflict, Process Conflict, and Relationship Conflict); (3) We aim to determine how different types of teacher-student conflict affect students’ creativity.

Theoretical background and literature review

2.1. Conflict theory and creativity

Previous research mainly focused on teacher-student relationship conflict [17, 24], but seldomly mentioned other types of teacher-student conflict. Several studies consistently concluded that relationship conflict is not conducive to students' creativity [24, 25]. Nevertheless, are all types of conflicts negative? Therefore, this paper will research all types of conflict in one theoretical model to explore their effects.

Creativity is a critical aspect of addressing sustainability [26, 27], it is the key skill to achieve sustainable development and foster more sustainable societies [28, 29]. Creativity assists us in gaining clarity of vision and accessing the breakthrough thinking required for sustainable development [30]. Researchers have scrutinized students’ creativity in relation to a variety of other constructs such as antecedents, mediators, and moderators [18, 31, 32]. However, sustainable development education is an ongoing process that emphasizes asking questions rather than accepting unchangeable answers [23]. Creativity in sustainable development education necessitates the development of personal consciousness, rather than absorbing prepackaged knowledge, conformity, and a limited intellectual scope [7]. Students’ creativity is the requirement of challenging behavior and critical thinking, especially in uncertain and ambiguous circumstances. In the classroom, teacher-student conflict is unforeseeable and uncontrollable. Therefore, it is crucial to investigate the relationship between teacher-student conflict and students’ creativity.

Point 2: In this case, the question also arises as to the employed research instrument for assessment of sustainable creativity. The impression is that the questionnaire that is used is intended simply to measure creativity, without considering its specific type, namely sustainable creativity. The connectedness of this instrument and sustainable creativity needs to be substantiated.

Response 2:

Based on the changes of response 1. The original creativity instrument was suitable for the concept of creativity, and we only changed the expression in the method section.

Revised Contents:

4.2. Measurement instruments

In the Students’ creativity:

It was assessed by the Creative Achievement Questionnaire [46] which concluded 10 domains with 61 questions, such as visual arts, music, and inventions. A sample item is “I wrote an original short story or poem for a class”.

Point3: In a general sense, the results of the article are relevant and interesting, but they are more reflective of the educational impact and links; therefore, the connectedness of this publication and the journal’s philosophy and ideas; i.e., Good-Health and Well-Being, Education and Social Inclusion, is questionable. The part of the discussion should be supplemented by proving that the obtained results pertain to these phenomena and how they relate to them.

Response 3:

We apologize for omitting this critical section and added the inclusions at the end of the research.

Revised Contents:

7. Conclusions

The findings of this study show how various types of conflicts impact students' creativity. Previous studies have shown that teachers focus on resolving or avoiding the conflict and their attitudes are not inclusive when challenged with a conflict. However, in light of the positive aspects of understanding conflict and processing conflict, teacher beliefs about conflict inclusion should be strengthened. This will not only aid in the occurrence of more beneficial conflicts, but will also benefit students' creativity. The teachers' inclusion should not be overlooked because it reveals the negative side of relationship conflict. Teachers' tolerance and concern for students can contribute to the enhancement of the interpersonal relationship between teachers and students, which can reduce the negative effect of relationship conflict.

Reviewer 2 Report

"The hypotheses 15 were supported by using data from questionnaires completed by 2,009 students at 17 American 16 universities".

The results are interesting and - for further research- it would be interesting to investigate the same hypothesis in European universities also.

Author Response

We appreciate your careful reading of our manuscript and the constructive comments. We apologized for omitting these critical problems and have made all revisions in response to your valuable suggestions.

Point1:

"The hypotheses 15 were supported by using data from questionnaires completed by 2,009 students at 17 American 16 universities".

The results are interesting and - for further research- it would be interesting to investigate the same hypothesis in European universities also.

Response1: We added this idea at the end of the Discussion section.

Revised Contents: We would also welcome further studies on the impact of teacher-student conflict on teachers’ creativity, and on how other students who are not directly involved in the conflict are motivated. And it would be interesting to investigate the same hypothesis in European universities as well.

Reviewer 3 Report

The article meets all the requirements, being a study that has 2027 participants.

The authors report the results of a descriptive study, correlations between factor scores, and a Hierarchical regression analysis. They could have presented other results that would have been of interest to the scientific community.

It would be necessary for the authors to update the reviewed sources including international works from the last 5 years, especially from the last 2 years, both in the introduction and in the discussion.

Author Response

We appreciate your careful reading of our manuscript and the constructive comments. We apologized for omitting these critical problems and have made all revisions in response to your valuable suggestions.

Point1: It would be necessary for the authors to update the reviewed sources including international works from the last 5 years, especially from the last 2 years, both in the introduction and in the discussion.

Response1: We changed and added new references both in the introduction and in the discussion, and they are all published in the last 2 years.

Revised Contents:

3. Kanzola, A.-M.; Petrakis, P. E., Τhe Sustainability of Creativity. Sustainability 2021, 13, (5), 2776.

4. Novikova, I. A.; Berisha, N. S.; Novikov, A. L.; Shlyakhta, D. A., Creativity and Personality Traits as Foreign Language Acquisition Predictors in University Linguistics Students. Behavioral Sciences 2020, 10, (1), 35.

8. Fredagsvik, M. S., The challenge of supporting creativity in problem-solving projects in science: a study of teachers’ conversational practices with students. Research in Science & Technological Education 2021, 1-17.

9. D'Souza, R., What characterises creativity in narrative writing, and how do we assess it? Research findings from a systematic literature search. Thinking Skills and Creativity 2021, 42, 100949.

12. Jiménez, T. I.; Moreno-Ruiz, D.; Estévez, E.; Callejas-Jerónimo, J. E.; López-Crespo, G.; Valdivia-Salas, S., Academic competence, teacher–student relationship, and violence and victimisation in adolescents: The classroom climate as a mediator. International journal of environmental research and public health 2021, 18, (3), 1163.

13. de Ruiter, J. A.; Poorthuis, A. M.; Koomen, H. M., Teachers’ emotional labor in response to daily events with individual students: The role of teacher–student relationship quality. Teaching and Teacher Education 2021, 107, 103467.

19. Zhao, G.; Xu, X.; Dye, D.; Rivera-Díaz-del-Castillo, P. E. J.; Petrinic, N., Facile route to implement transformation strengthening in titanium alloys. Scripta Materialia 2022, 208.

20. Chen, X.; He, J.; Fan, X., Relationships between openness to experience, cognitive flexibility, self-esteem, and creativity among bilingual college students in the U.S. International Journal of Bilingual Education and Bilingualism 2022, 25, (1), 342-354.

29. Bassachs, M.; Cañabate, D.; Serra, T.; Colomer, J., Interdisciplinary Cooperative Educational Approaches to Foster Knowledge and Competences for Sustainable Development. Sustainability 2020, 12, (20), 8624.

47. Civitillo, S.; Göbel, K.; Preusche, Z.; Jugert, P., Disentangling the effects of perceived personal and group ethnic discrimination among secondary school students: The protective role of teacher–student relationship quality and school climate. New Directions for Child and Adolescent Development 2021.

48. Roslyne Wilkinson, H.; Jones Bartoli, A., Antisocial behaviour and teacher–student relationship quality: The role of emotion‐related abilities and callous–unemotional traits. British Journal of Educational Psychology 2021, 91, (1), 482-499.

49. Roper, I.; Higgins, P., Hidden in plain sight? The human resource management practitioner's role in dealing with     workplace conflict as a source of organisational–professional power. Human Resource Management Journal 2020, 30, (4), 508-524.

50. Kurbonalievna, I. G.; Adxamovna, B. G., Innovative solutions for effective conflict resolution in higher education institutions. South Asian Journal Of Marketing & Management Research 2021, 11, (6), 33-37.

51. Todorova, G.; Goh, K. T.; Weingart, L. R., The effects of conflict type and conflict expression intensity on conflict management. International Journal of Conflict Management 2021.

52. Ma, L.; Liu, J.; Li, B., The association between teacher‐student relationship and academic achievement: The moderating effect of parental involvement. Psychology in the Schools 2022, 59, (2), 281-296.

Round 2

Reviewer 3 Report

After the review carried out by the authors, I consider that the article can be published in the journal